# Availability and Access to Orphan Drugs for Rare Cancers in Bulgaria: Analysis of Delays and Public Expenditures

**DOI:** 10.3390/cancers16081489

**Published:** 2024-04-12

**Authors:** Kostadin Kostadinov, Ivelina Popova-Sotirova, Yuliyana Marinova, Nina Musurlieva, Georgi Iskrov, Rumen Stefanov

**Affiliations:** 1Department of Social Medicine and Public Health, Faculty of Public Health, Medical University of Plovdiv, 4002 Plovdiv, Bulgaria; ivelina.popova@mu-plovdiv.bg (I.P.-S.); yuliyana.marinova@mu-plovdiv.bg (Y.M.); nina.musurlieva@mu-plovdiv.bg (N.M.); georgi.iskrov@mu-plovdiv.bg (G.I.); rumen.stefanov@mu-plovdiv.bg (R.S.); 2Institute for Rare Diseases, 4023 Plovdiv, Bulgaria

**Keywords:** rare cancers, orphan drugs, Bulgaria, cancer costs, health inequalities

## Abstract

**Simple Summary:**

In this study, we investigate the availability and access to orphan drugs for rare cancers in Bulgaria, aiming to address the urgent need for improved treatment access for this vulnerable population. By comparing data from European and national sources, we aimed to assess the availability, delays, and budgetary impact of these drugs. Our findings reveal significant disparities in their access and highlight the pressing need for targeted policies to address these inequalities. This research contributes valuable insights into the challenges faced by rare cancer patients and we call for focused efforts at both the European and national levels to ensure equitable access to treatment.

**Abstract:**

Rare cancers are defined by an annual incidence of fewer than 6 per 100,000. Bearing similarities to rare diseases, they are associated with substantial health inequalities due to diagnostic complexity and delayed access to innovative therapies. This situation is further aggravated in Southeastern European countries like Bulgaria, where limited public resources and expertise underscore the need for additional policy and translational research on rare cancers. This study aimed to explore the availability and access to orphan drugs for rare cancers in Bulgaria for the period of 2020–2023. We cross-compared data from both the European Union and national public sources to evaluate the number of available and accessible orphan drugs for rare cancers, the delay from market authorization to reimbursement, the dynamics of public expenditures, and regional disparities in access across the country. We juxtaposed the main characteristics of oncological and non-oncological orphan drugs as well. Only 15 out of 50 oncological orphan drugs that were authorized by the European Medicine Agency were accessible for rare cancer patients in Bulgaria. The median delay between market authorization and inclusion in the Bulgarian Positive Drug List was 760 days. The total expenditures for all orphan drugs for rare cancers amounted to EUR 74,353,493 from 2020 to 2023. The budgetary impact of this group rose from 0.24% to 3.77% of total public medicinal product expenditures for the study period. Rare cancer patients represent a vulnerable population that often faces limited to no access to treatment. We call for targeted European and national policies to address this major inequality.

## 1. Introduction

### 1.1. Background

Rare cancers are typically defined by an annual incidence of fewer than 6 per 100,000 [1]. While this threshold is widely accepted within the scientific community [2,3] and the regulatory frameworks in certain countries [4], a common EU legal definition has yet to be adopted [5]. Rare cancers encompass a diverse group of diseases, accounting for 24% of all cancer cases in Europe [6]. However, the affected patients share common health inequities, leading to a significantly lower 5-year survival rate compared to common cancer types [7]. Similar to rare diseases [8], rare cancers are characterized by health discrimination [9], scarce information [10,11], limited research opportunities [12], lack of targeted preventive policies [13], difficulties in accurate diagnosis [14], and fragmented clinical management [15,16,17,18,19]. While these public health problems are evident at the member state level, it is also observed that Eastern European countries exhibit significantly lower 5-year survival rates for rare cancers compared to their Western European counterparts [6]. Such findings underline the crucial impact of healthcare system organization and accessibility to innovative therapies.

### 1.2. Rare Cancer Policies

Two significant groups of EU policies have focused on the challenges of rare cancers. First, Europe’s Beating Cancer Plan (EBCP) [20] included rare cancers as one of its key priorities. Aiming to meet the increasing demands for cancer prevention, effective screening, and treatment, several of the proposed EBCP initiatives offer investments in rare cancer research and innovation, tackle the lack of therapeutic protocols, and address the limited access to innovative therapies. However, the EBCP does not explicitly provide a standardized healthcare organizational model for the management of rare cancers. Thus, the two main approaches—centralized “peer to peer” [21] and decentralized “hub and spoke” [22]—remain equally valid alternatives considering the member states’ healthcare system characteristics. Still, no sufficient evidence exists to support the preferred model among member states [13,22,23].

The second main group of policies considers rare cancers as rare diseases with an oncological manifestation. Therefore, rare cancer stakeholders could also benefit from the already-developed rare initiatives such as the Orphan Drug Legislation [24] and the European Reference Networks (ERNs) [25,26]. The orphan drug designation tackles small-market barriers by providing economic incentives and facilitating the regulatory approval process. However, drug policy should also promote equity and fairness, which are crucial in the case of rare cancers, for which classical randomized clinical trials can rarely reach significant results [27,28,29]. The lack of real-world post-authorization data and RCT outcome generalizability further increases therapeutic uncertainty [30]. Despite the number of authorized innovative medicinal products [31], several studies indicate potential orphan drug market failures. The substantial increase in oncological indications outweighs all other rare diseases, leading to market substitution [19,32,33,34]. This expansion is partly attributed to the “salami slicing” phenomenon, where the patient population is intentionally divided into smaller subgroups based on cancer stage and positive genetic or immunological markers [19,35,36,37]. While this approach narrowly focuses on patient eligibility, certain orphan cancer drugs may also be used in non-rare or multiple rare conditions, shifting the market dynamics to resemble conventional medical therapies [38,39,40,41,42]. Yet, numerous rare cancers, especially in children, continue to face significant unmet health needs [43].

### 1.3. Access to Orphan Cancer Therapies in the EU and Bulgaria

Access to orphan cancer drugs in the EU faces challenges in two main stages. While the Orphan Drug Regulation promotes research and development and sets up conditions for market authorization by the European Medicines Agency (EMA), pricing and reimbursement decisions remain at the discretion of member states [44,45]. The latter usually involves health technology assessment (HTA) and price negotiations. However, the economic discrepancies across the EU result in significant heterogeneity in reimbursement decisions. Thus, while some countries base the reimbursement decision on specific HTA criteria for oncological or rare therapies, others rely on budgetary impact, multicriteria analysis, or higher cost-effectiveness thresholds [46,47].

The Bulgarian case is no exception. Patient access to innovative cancer drugs includes a complex procedure in several stages [48]. The process is initiated by market authorization holders who apply to the National Council on Prices and Reimbursement of Medicinal Products (NCPRMP) for inclusion in the Positive Drug List (PDL) and public fund reimbursement. To be included in the PDL, an innovative cancer drug must already have been funded in five out of the seventeen reference countries. In the case of orphan drugs, this reference list extends to all EU member states [49]. The pricing of the product is based on an external reference list. The maximum product price is set to be no higher than the lowest price among a list of 10 reference countries [49]. In addition, market holders submitting applications for public reimbursement are required to contract a discount agreement with the single public payer, the National Health Insurance Fund (NHIF). The minimum discount is set at 10% of the official selling list price. However, the exact discount is confidential. When the maximum price and discount agreement are set, the NCPRMP conducts an HTA assessment based on cost-effectiveness analysis (CEA) and budget impact analysis (BIA). Legally, this stage is supported by evidence from HTA reports from the UK, France, Germany, and Sweden or a joint clinical assessment report in relation to Regulation (EU) 2021/2282 [50]. Finally, the NHIF is responsible for establishing coverage and distribution methods. Hospital pharmacies handle the distribution of all drugs listed and eligible for reimbursement. Hospitals procure these drugs either through direct contracts or public procurement, with decentralized purchasing processes. The NFIF reimburses hospitals monthly for cancer drug payments, with a maximum price deviation enforced nationally. Costs exceeding this deviation are reimbursed by market authorization holders, as are expenses exceeding the allocated NHIF budget [51].

### 1.4. Rare Cancer Policies in Bulgaria

The Bulgarian Cancer Control Plan (BCCP) [52], similar to the EBCP, also includes rare cancers as one of its priorities. The plan identifies the lack of specific HTA criteria in the reimbursement framework as one of the main barriers to access to innovative therapies. However, the BCCP does not set goals based on Bulgarian orphan drug market practices, real-world data on budgetary impact, or existing regional differences in access. This is particularly important in the context of emerging needs for policy research and evidence-based decision-making for rare cancers [5,53].

The primary objective of this research is to conduct a thorough analysis of the orphan cancer drug market in Bulgaria, juxtaposing it against innovative non-orphan cancer drugs. Moreover, this study provides a comprehensive analysis of the regional differences in access to orphan cancer drugs and assesses their budgetary impact from a healthcare perspective. The study results provide insights that can inform evidence-based policy decisions concerning rare cancers at both the national and EU levels.

## 2. Materials and Methods

### 2.1. Study Design

This study is a retrospective analysis of orphan drugs (ODs) in the oncology market in Bulgaria between July 2020 and September 2023. The start of the study period was selected as the first month for which open data on cancer OD expenditures at the hospital level were available.

Eligible drugs were identified using the EMA’s European Public Assessment Reports (EPARs) database. This study focused on cancer drugs used in human medicine, as defined by the Anatomical Therapeutic Chemical (ATC) classification system and the International Classification of Diseases (ICD) therapeutic indication. The market (trade) name of the drug was used as a unique identifier to match the EMA database with all relevant data sources. Only drugs with active marketing authorization were included in the analysis. The cost data were extracted only for medications listed in the Bulgarian Positive Drug List (PDL) and eligible for reimbursement.

Four main aspects of the cancer OD market were analyzed: (1) the number of reimbursed cancer ODs and their access delay, (2) the public expenditures of reimbursed cancer ODs and their budgetary impact, (3) the regional differences in access and expenditures for cancer ODs, and (4) the comparative profile of market authorization holders. The estimated expenditures, delays, and budgetary impact were compared to the corresponding data for innovative non-orphan cancer drugs. The cancer medicines without OD were selected based on the match between the EMA database and the Bulgarian PDL. The study data as well as the performed analysis are publicly available on the project’s GitHub repository https://github.com/kostadinoff/Availability-and-access-to-orphan-drugs-for-rare-cancers-in-Bulgaria, created on 13 March 2024.

### 2.2. Definitions

In this study, the term “access” is defined as the ability to receive timely and fully funded medical treatment. Within this framework, drugs approved and registered by the EU as ODs are considered “available”, but they are only deemed “accessible” when they are integrated into the public health system through an effective reimbursement scheme [54]. Thus, the term “access delay” is used to describe the time interval between the EMA market authorization and the first NHIF expenditure for a given cancer OD. The overall delay was divided into two main components: “external delay” and “internal delay”. The term “external delay” is used to describe the time interval between the EMA market authorization and the inclusion of a specific cancer OD in the PDL. The term “internal delay” is used to describe the time interval between the inclusion of a specific cancer OD in the PDL and the first NHIF expenditure.

### 2.3. Data Collection

Data were collected on 3 January 2024. Four main data sources were used for this study (Table 1): (1) NHIF data, (2) EMA’s EPARs database, (3) NCPRMP’s PDL database, and (4) NHIF annual budget reports.

The EPARs database was downloaded from the EMA website. The dataset was filtered by the ATC as well as the ICD of the therapeutic indication to include only cancer drugs used in human medicine. The variables of interest included the product name, ATC code, therapeutic indication, orphan designation status, and the date of market authorization. Only products with active marketing authorization were included.

Cost data were extracted from the NHIF website. The scraping resulted in 43 Excel data files spanning from July 2020 to September 2023. Each file underwent a comprehensive review to ensure data integrity and consistency. The variables of interest included product name, number of treated patients, ATC, INN, region, hospital, and ICD code. The data processing was conducted using R [55] and the tidyverse package [56]. This process involved consolidating the data files into a unified time series database with one row per product. Additional steps included standardizing the data format and metadata checks. Subsequently, the database was converted into a CSV file for further analysis.

PDL data were downloaded from the Open Data Portal of the Republic of Bulgaria. The database was filtered by the market name matched in the EPARs database. The variables of interest included the reimbursement status and the date of inclusion in the PDL.

Finally, the NHIF annual budget reports were obtained from the NHIF website. The expenses related to all medicinal treatments were extracted from the reports covering the years 2020–2023.

### 2.4. Data Analysis

The data analysis was conducted utilizing R version 4.3.2 [55]. Categorical variables were summarized with counts and percentages, while for continuous variables, median, minimum, and maximum values were used. Bootstrap simulation with replacement was employed to estimate the ratio or difference between two medians, along with generating bootstrapped 95% confidence intervals. Additionally, the Mann–Whitney U test was utilized to assess differences between the two groups. Monthly and yearly growth rates were calculated using the following formula: $GR=\frac{y_{t}-y_{t-1}}{y_{t-1}}\times 100$. All cost data were converted from Bulgarian currency (BGN) to the Euro (EUR) using the official fixed exchange rate of 1 EUR = 1.95583 BGN. Statistical significance was set at *p* < 0.05. Data visualization was performed using the ggplot2 package [57]. Trend lines were fitted using the LOESS method. The budgetary impact of cancer ODs was estimated as a proportion of the total cancer treatment expenditures (CTE) and the National Health Insurance Fund Medicines Expenditure (NHIF ME). In the cases where the data did not cover a full 12 months, the NHIF ME was estimated by dividing the annual budget by 12 and multiplying it by the number of months covered in missing data periods. The median monthly expenditure (MME) per drug was calculated by dividing the total expenditures by the number of months in the study period.

## 3. Results

### 3.1. Number of Available Cancer ODs and Their Access Delay

On 30 September 2023, the analysis of the EPARs list revealed 50 ODs with an active centralized EU marketing authorization. Subsequently, each of these therapies underwent evaluation within the Bulgarian PDL database, resulting in 15 registered and priced products (30%) eligible for reimbursement (Table 2). The median increase rate for the number of cancer ODs in the PDL was 7% (Figure 1A), which was 3.4 times slower compared to the increase rate of EMA authorization (24%).

The median external delay (MED) for cancer ODs, defined as the time interval between EMA authorization and PDL inclusion, was 760 days. The leaders in this category were Imnovid^®^ (2331 days) and Vyxeos Liposomal^®^ (1574 days). Rydapt^®^ exhibited the shortest MED (235 days). The MED for all 157 non-orphan cancer drugs matched with the EPARs database was 793 days. The product with the longest external delay was Targretin^®^ (6812 days), while the shortest was observed for Kanjinti^®^ and Inflectra^®^ (both 78 days). The difference in MEDs between orphan and non-orphan cancer drugs was not statistically significant (bootstrap estimated median difference = −33 days; 95% BCa [−222.12; 508.88]).

At the beginning of the study period (July 2020), funding had already been provided for 10 of the 15 cancer ODs. Subsequently, between 2021 and 2023, five new cancer ODs were initially funded. For this later group (Figure 1C,D), the estimated median time to reimbursement was 988 days (min—351 days; max—1743 days). The median internal delay (from PDL inclusion to the initial NHIF expenditure) was 469 days (min: 2 days; max: 518 days).

### 3.2. Expenditures of Accessible OD Cancer Drugs and Their Budgetary Impact

The total estimated expenditures for all cancer ODs during the study period (July 2020–September 2023) reached EUR 74,353,493 (Figure 2A). Over time, a consistent upward trend was observed, with a median growth rate of 4.86% per month. The lowest monthly expenditures were recorded in September 2020 at EUR 1,157,494, while the highest peak occurred in August 2023 (EUR 3,655,748). Two significant outliers were observed in the trend line: one in January 2023 with a growth rate of −50.5% and another in February 2021 with a growth rate of +88.8%.

The upward trend in February 2021 was predominantly attributed to increased costs of Nexavar^®^, indicated for three types of malignancies (hepatocellular carcinoma, renal cell carcinoma, and differentiated thyroid carcinoma). In February 2021, the number of patients treated with Nexavar^®^ increased 7.9 times compared to the previous month (102 vs. 13), resulting in additional expenditures of EUR 256,788. Another contributing factor was the expansion of the regions where cancer ODs were provided (eight in January 2021 vs. fourteen in February 2021).

On the other hand, in January 2023, there was a sharp decrease in the number of patients and the corresponding expenditures for 13 out of the 14 accessible cancer ODs at that moment. The most significant drop was observed for Darzalex^®^ (from EUR 840,812 for 93 patients in December 2022 to EUR 184,763 for 20 patients in January 2023). The only cancer OD that did not experience a decrease was Nexavar^®^ (from EUR 25,666 for nine patients in December 2022 to EUR 308,634 for one hundred and six patients in January 2023). In January 2023, no public costs were recorded for five cancer ODs (Blincyto^®^, Mylotarg^®^, Onivyde Pegylated Liposomal^®^, Polivy^®^, and Xospata^®^).

The primary contributor to the overall cancer OD expenditures was Darzalex^®^ (Figure 2B). This product was approved for previously treated or newly diagnosed multiple myeloma. Darzalex^®^ was registered in the PDL on 20 December 2018 and reimbursed every month throughout the study period, comprising 30% of all cancer OD expenditures (EUR 22,299,119). In contrast, Vyxeos Liposomal^®^, included in the PLD on 14 December 2022, only accounted for 0.1% (EUR 73,100). Moreover, due to a combined internal and external delay, public costs for Vyxeos Liposomal^®^ were identified in only three months (June, August, and September 2023) and were provided in only three out of the forty-six cancer treatment hospitals (two in the city of Sofia and one in the city of Varna).

The collective expenses for all cancer medicines reached EUR 1,313,676,147. Among the 248 cancer drugs covered, 27 (two ODs and twenty-five non-ODs) constituted 80% of the total expenditure. In the non-OD category, Keytruda^®^ emerged as the leader, contributing to 21% of all non-OD expenditures (EUR 259,827,300), followed by Xtandi^®^ (6.7%–EUR 82,695,905) and Opdivo^®^ (EUR 5.3%–65,499,689). The median total cost for the 15 cancer ODs was EUR 2,156,224, while for the remaining 233 non-orphan cancer drugs, it was EUR 235,535 (Figure 2D). The results of the Mann–Whitney U test indicated a significant difference between the two groups (U = 845, *p* = 0.0008, rank-biserial correlation = −0.52 [−0.70, −0.26]). The Bootstrap simulation revealed that the median total expenditure for the OD group was 9.15 times higher than the non-OD group (4000-fold, 95% BCa 4.871–20.836).

The yearly budgetary impact of cancer ODs (July 2020–September 2023) is outlined in Table 3. A rising trend of 3.53 percentage points was observed in the proportion of cancer ODs within NHIF medical expenditures, escalating from 0.24% in 2020 to 3.77% in 2023. Similarly, the percentage of all cancer drugs rose by 1.96 percentage points, progressing from 4.75% in 2020 to 6.71% in 2023. Notably, between 2021 and 2023, the increase in the proportion of cancer ODs within all cancer medicines expenditures surpassed the increase in NHIF medical expenditures.

The median monthly expenditure (MME) for cancer ODs in the study period was EUR 1,580,744 (min: EUR 1,157,494; max: EUR 3,655,748). In contrast, the estimate for non-orphan cancer drugs was EUR 29,803,136 (min: EUR 24,068,096; max: EUR 47,242,005). A comparison between the two groups incorporating the median monthly number of patients and the median monthly expenditure per patient is presented in Table 3. Overall, the median monthly expenditure per patient in the OD cancer group was EUR 5427 EUR (minimum: EUR 4435; maximum: EUR 7326). In contrast, for non-orphan cancer drugs, the estimate was EUR 1300 (minimum: EUR 1142; maximum: EUR 1534). The ratio between the two indicators was 4.17 (Figure 2D).

### 3.3. Regional Differences in Access and Expenditures for Cancer ODs

Reimbursement for cancer medicinal treatment was allocated in 16 out of the 28 regions in Bulgaria (57.14%), with cancer ODs being administered in 15 of them (93%), with Sofia Province being the sole exception (Table 4). The capital city, Sofia, recorded the highest overall cancer OD expenditures, amounting to EUR 43,675,774 (58.74% of all cancer OD expenditures), aligning with the highest overall MMPT (176). Additionally, the capital had the highest number of cancer OD hospital providers (n = 16)—2.6 times more than the second-place region (Plovdiv, n = 6). The lowest total cancer OD expenditures were recorded in the regions of Gabrovo (EUR 7164) and Veliko Tarnovo (EUR 35,168). The lowest median monthly expenditures and the median monthly expenditure per patient were observed in Veliko Tarnovo (EUR 1610, min: EUR 1354, max: EUR 3198) where only one hospital provided two cancer ODs (Figure 3A).

At least one cancer OD was administrated in 41 out of the total 46 hospitals included in the database (Figure 3B). Nexavar^®^ emerged as the most widely utilized medicine, being administered in 36 hospitals (78.3%) across 15 regions. This was followed by Onivyde^®^, which was dispensed in 29 hospitals (63%) spanning 13 regions. Vyxeos Liposomal^®^ and Besponsa^®^ were the least regionally accessible cancer ODs, each one being administered in only three and four hospitals, respectively.

The regional distribution of cancer medicine costs, encompassing both orphan and non-orphan-designated products, is presented in Figure 3C,D. The city of Sofia recorded the highest proportions for cancer medicines expenditures, accounting for 59% in the OD group and 42% in the non-OD group. At the bottom of the list were the regions of Gabrovo, exhibiting a share of 0.01% of all cancer OD expenditures, and Dobrich, with the lowest proportion of non-orphan cancer drug expenditures, at 0.12%. The median ratio between orphan- and non-orphan-designated cancer drug shares was 0.226 (min: 0.015; max: 1.78). Three regions (Pleven, Varna, and the city of Sofia) exhibited a ratio greater than 1, indicating that the OD market shares surpassed the non-orphan ones in these regions.

### 3.4. Comparative Profile of Market Authorization Holders

The 15 accessible cancer ODs in the Bulgarian market were provided by 12 market authorization holders (MAHs). The leading company in terms of market size was Janssen-Cilag International N.V. (Darzalex^®^). This is followed by Takeda, whose two cancer ODs, Adcetris^®^ and Ninlaro^®^, collectively accounted for 26.94% of all cancer OD expenditures. On the contrary, Novartis Pharma AG ranked lowest, with Rydapt^®^ sharing only 0.01% of all cancer OD expenditures. However, other Novartis products (Afinitor^®^, Kisqali^®^, Mekinist^®^, Tafinlar^®^, Tasigna^®^, Tyverb^®^, Piqray^®^, and Votrient^®^) contributed notably to the non-orphan cancer drug market (8.54%).

The market share of each MAH for both orphan and non-orphan cancer drug products in Bulgaria is presented in Figure 4. To assess the oncology market profile, the differences between the OD and non-orphan cancer drug expenditure shares were calculated for each company and then standardized using the z-score transformation. Companies with a z-score above 1 were Janssen-Cilag International N.V. (1.75) and Takeda (2.11), indicating a prominent OD profile. Conversely, the company with the lowest z-score was Roche (−1.07), suggesting a preference for a non-orphan market profile.

## 4. Discussion

One of the main findings of this study emphasizes the restricted access to cancer ODs in Bulgaria. Several obstacles hinder this process. Initially, many drugs fail to provide evidence of a clinical benefit sufficient to meet market authorization criteria. While this can be attributed to the complexity of the cancer disease, it may also be influenced by the clinical trial design [27]. Still, similar clinical outcomes lead to different authorization decisions. The EMA approval probability of anticancer drugs stays lower than the average for all other medicinal products and lags behind the FDA in the USA [58]. Still, the approved cancer ODs in the EU often rely on surrogate endpoint data, such as progression-free survival (PFS) or objective response rates from single-arm or observational studies [59]. This hope-driven approach [60], as found in our study, speeds up the authorization process (24% of median increase rate in the number of cancer ODs) but also transfers the patient delay to the post-authorization phase due to the unaddressed clinical uncertainty [61].

While this first barrier affects all EU member states, the post-authorization delay disproportionately discriminates against economically disadvantaged countries in Eastern and Southeastern Europe [62]. This is evident in the Bulgarian case, where the median time to PDL inclusion for all 15 out of 50 authorized cancer ODs was 760 days. However, it should also be underlined that cancer ODs experience a comparable median delay to innovative but non-orphan-designated cancer medicines (793 days). Similar results regarding the Bulgarian market were revealed in previous periods [63,64]. In contrast, Post et al. demonstrated a significantly shorter median time for the reimbursement of innovative cancer drugs in Germany, France, the UK, the Netherlands, Belgium, Norway, and Switzerland (407 days) [65]. Parallel to our findings, the authors observed that orphan status was not statistically associated with accelerated access.

Many factors contribute to post-authorization delay, including clinical uncertainty, pharmaceutical company strategy, HTA methodology, and economic disparities [66]. In their nature, these factors align rare cancers more closely with other oncological conditions than with rare genetic disorders. Moreover, for many rare cancers, treatment alternatives often overlap with those available for common cancer types [67,68]. However, the OD price tags, often higher than innovative non-orphan products, distinguish significantly rare cancers [44,69]. This is evident in our study, where the median per-product cost for all 15 cancer ODs was EUR 2,156,224, while for the remaining 233 non-orphan cancer drugs, it was EUR 235,535. The high price tag is often considered a hurdle for patient access [62,66,70]. While some decision-makers may be prompted to overlook this issue, the budgetary impact of cancer ODs should also be considered [71,72]. In our study, the budgetary impact estimated as a proportion of all medicinal expenditures of the NHIF remained under 4% for all years in the study period. Regarding the share of cancer OD expenditures in the total cancer treatment expenditures, our results revealed a 41% increase (from 4.75% in 2020 to 6.71% in 2023). This trend is aligned with the findings of a previous study [73], which reported an increase in the share of cancer treatment expenditures in the NHIF budget. The reason for this was explained by the increased trend in oncological morbidity and the inclusion of new expensive drugs in the PDL, paid for with public funds. Still, this share remains lower than expected considering that almost one-quarter of all cancers are rare [6]. This discrepancy, we believe, is attributed to the limited treatment options accessible on the Bulgarian market.

The budgetary impact of rare cancers may be an unreliable indicator. Reimbursement of a new OD cancer treatment is often associated with collateral expenditures, as many cancer ODs are used in combination with other therapies, require previous treatment with other drugs, or require additional diagnostic procedures [74,75]. Thus, while the incremental cost-effectiveness ratio of rare cancer therapies does not differ significantly from that of non-rare cancer therapies, the budgetary impact of cancer ODs should be considered in the context of the entire treatment pathway [76].

The regional differences in access and expenditure for cancer ODs are another important finding of this study. The capital city, Sofia, recorded the highest overall cancer OD expenditures, aligning with the highest overall median monthly per-patient expenditures. This was not a surprising result considering that Sofia is also the most populated region in Bulgaria. However, the difference in the regional distribution of cancer OD vs. non-orphan cancer drug expenditures provides insights regarding the healthcare model for rare cancers. In three regions (Pleven, Varna, and Sofia), expenditures for cancer ODs surpassed those for non-orphan cancer drugs. This finding is indicative of a centralized approach resembling the “peer-to-peer” model, where only a few specialized hospitals handle the treatment of the vast majority of rare cancer patients [21]. Furthermore, the ODs provided in those regions outnumber the remaining regions. This is in line with the results of a recent study by Vancoppenolle et al. [77], in which patient access to specialized hospitals was significantly accelerated. However, several studies contradict the centralized approach, arguing that the “hub and spoke” model is more appropriate. This model assumes that the treatment of rare cancers should be provided in proximity under the supervision of a specialized center. Such an approach may not only reduce the travel burden for patients and their families but also combat the differences in regional survival rates [22,78,79]. However, further research is needed to assess the preferred model in Bulgaria among patients, healthcare professionals, and decision-makers.

An additional challenge faced by already approved and reimbursed cancer ODs involves market shortages and out-of-pocket payments. While this study does not delve into the latter aspect, it is worth noting that market shortages, though more common for generic non-orphan cancer drugs, can also affect rare cancer patients. In January 2023, our study found no public costs associated with five cancer ODs. A reduction in expenses was also observed for several non-orphan cancer drugs. While the precise reasons for this occurrence are unclear, two potential factors should be highlighted. First, supply shortages have been identified as a significant barrier, particularly in certain Eastern European countries [80,81]. However, political factors, leading to delays in NHIF budget approval, may also play a role.

Market competition is regarded as the primary driver for price reduction and innovation in oncology [82]. Our study also provides data on the competitive landscape in the oncology market in Bulgaria. For all 15 cancer ODs, 12 MAHs were identified, with Janssen-Cilag International N.V. leading in terms of market size. However, only two companies have been identified as the primary contributors to the cancer OD market: Janssen-Cilag International N.V. (Darzalex^®^) and Takeda (Adcetris^®^ and Ninlaro^®^). The market profile of the other five preferred the non-orphan cancer drug market. Two companies (Wyeth Europa and Jazz Pharmaceuticals) were MAHs for cancer ODs only.

Our research indicates that the Bulgarian market for cancer ODs shares similarities with other innovative but non-orphan cancer medications. Access to all cancer drugs follows a similar regulatory pattern and challenges [83], resulting in comparable post-authorization delays. However, cancer ODs carry a significantly higher price, mostly attributed to the non-targeted oncological therapies and conventional chemotherapy included in the non-orphan category [84,85].

Similar to Bulgaria, innovative cancer drugs in neighboring Balkan countries are associated with a substantial increase in public expenditures [86,87,88]. Recent research also suggests that orphan status alone does not significantly predict the time to access. Factors such as health expenditure per capita, GDP, Mackenbach score of health policy performance, ESMO magnitude of clinical benefit scale scores, and market prices of medicines have a significantly higher impact on reimbursement delays [88,89].

Both orphan and non-orphan cancer drugs are associated with a high level of clinical uncertainty, which is often a problem in reimbursement decision-making [90]. The effectiveness of EU regulatory policies, such as conditional marketing authorization or adaptive pathways, often remains uncertain. Prolonged regulatory review processes could largely counteract the modest and non-significant positive impact of these incentives on clinical development timelines [90]. On the other hand, however, evidence also suggests that oncology drugs may unduly expand the early access tools designed for rare diseases, leading to a call for more stringent regulation of drugs with multiple indications [91]. All these factors cluster the cancer ODs more closely to the non-rare innovative cancer market than to the general OD market.

Our research also highlights a significant disparity between market authorization trends in the EU and access trends in Bulgaria, which has widened significantly over the study period. The current regulatory framework in Bulgaria lacks two critical legal mechanisms that could potentially help mitigate this gap.

First, the absence of a specific regulatory framework for patient engagement in the HTA process may result in suboptimal decisions regarding the coverage and reimbursement of orphan cancer medicines [92]. Patient engagement in HTA is vital for ensuring that the perspective of affected individuals and their caregivers is considered in the decision-making process. This factor has long been acknowledged as a powerful tool for the successful implementation of HTA policies [93].

Second, Bulgarian HTA legislation does not offer specific guidance for the assessment, appraisal, and reimbursement of testing practices, prognostic markers, and other components of personalized oncology medicine. Public funding often does not cover the companion diagnostic tests required for the administration of many orphan cancer drugs [94,95]. This barrier could impede the price negotiation process and contribute to access delays for both orphan and non-orphan cancer medicines [95].

## 5. Limitations

There are several important limitations that need to be considered when discussing the results of this study. First, the data collection process was organized by focusing on medicinal costs covered by the public funder, NHIF. Consequently, the study does not offer insights into out-of-pocket payments or the private insurance market in Bulgaria. Additionally, other treatment-related expenses, such as diagnostic procedures, surgery, or radiotherapy, were not considered. It is important to acknowledge that, while the study distinguishes between orphan and non-orphan cancer drugs, rare cancer patients may often receive off-label targeted treatment overlapping with non-orphan cancer drugs, standard chemotherapy, or palliative care. Indirect expenditures, such as travel costs and formal and non-formal care, were also not included. Thus, the estimated expenditures for cancer ODs should not be interpreted as the total costs of rare cancer treatment.

Second, this study’s timeframe was limited, potentially not allowing for a comprehensive understanding of the complete market dynamics. Furthermore, external political and market factors during the study period may also have influenced the results. Additionally, concerns about data quality arise as the NHIF database was designed for administrative purposes. Thus, applying it for research purposes required several data-processing steps. While the data were reviewed for consistency and integrity, data entry validation was not under the control of the authors.

Third, the dataset provided as an addition to this study includes the number of patients treated monthly at the hospital level. However, the data do not include information on the patient’s dynamics or patient identifiers. As a result, the incidence of rare cancers and clinical outcomes cannot be extrapolated.

Fourth, the estimated external delay was calculated based on the date of EMA market authorization and the date of inclusion in the PDL. However, two main factors that contribute to the external delay were not analyzed in this study. The first factor is the time interval between the EMA market authorization and the submission of the reimbursement dossier to the NCPRMP. This period is often prolonged due to the need for additional evidence or product funding approval in other EU countries. The second factor is the time interval between the submission of the reimbursement dossier to the NCPRMP and the inclusion of the product in the PDL. That period can be prolonged due to the administrative burden, the need for additional evidence, or the need for price negotiations.

Lastly, the study did not account for potential variations or administrative barriers in clinical practices across different healthcare providers and regions, which might impact the generalizability of the findings.

## 6. Conclusions

The findings of this study highlight the Bulgarian cancer OD market’s distinctive features, encompassing a limited number of accessible products and significant delays in access. Moreover, the research identifies substantial regional variations in both access and expenditures. The implications of these study outcomes extend beyond academic insights, offering valuable insights that can guide rare cancer policy development.

## Figures and Tables

**Figure 1 cancers-16-01489-f001:**
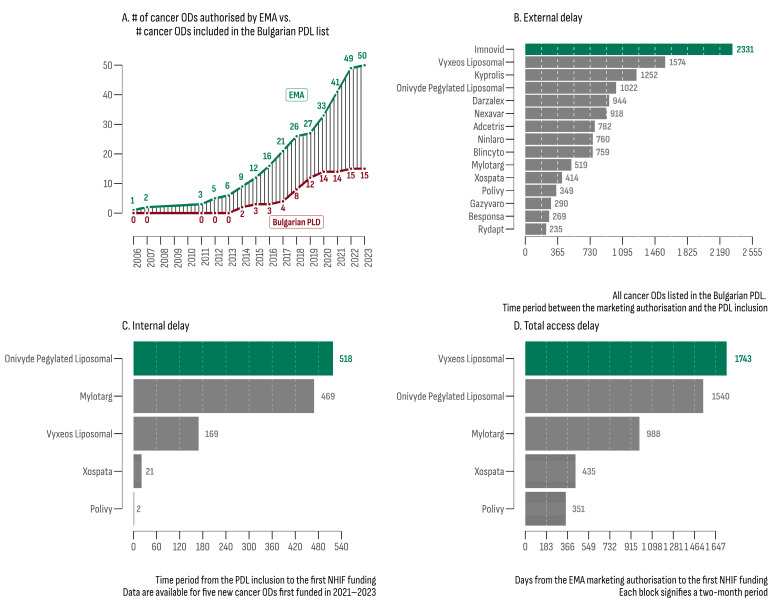
(**A**) Number of OD cancer drugs authorized by the EMA and their accessibility in Bulgarian PDL; (**B**) external delay between EMA authorization and inclusion in the PDL; (**C**) internal delay between PDL inclusion and first NHIF expenditure; (**D**) total delay between EMA authorization and first NHIF expenditure.

**Figure 2 cancers-16-01489-f002:**
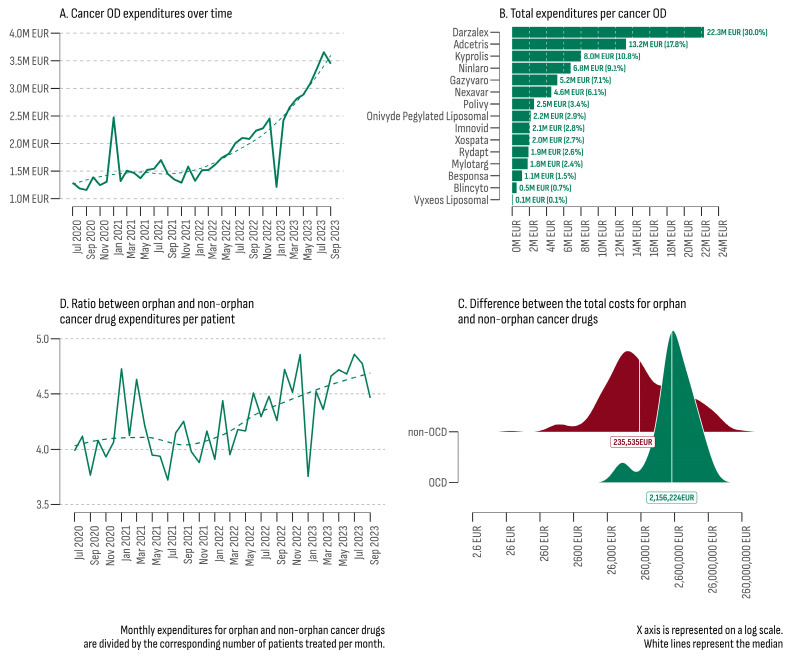
(**A**) Total cancer OD expenditures (July 2020–September 2023); (**B**) total expenditures per cancer OD; (**C**) difference between the total costs for orphan and non-orphan cancer drugs; (**D**) ratio between orphan and non-orphan cancer drug expenditures per patient (July 2020–September 2023).

**Figure 3 cancers-16-01489-f003:**
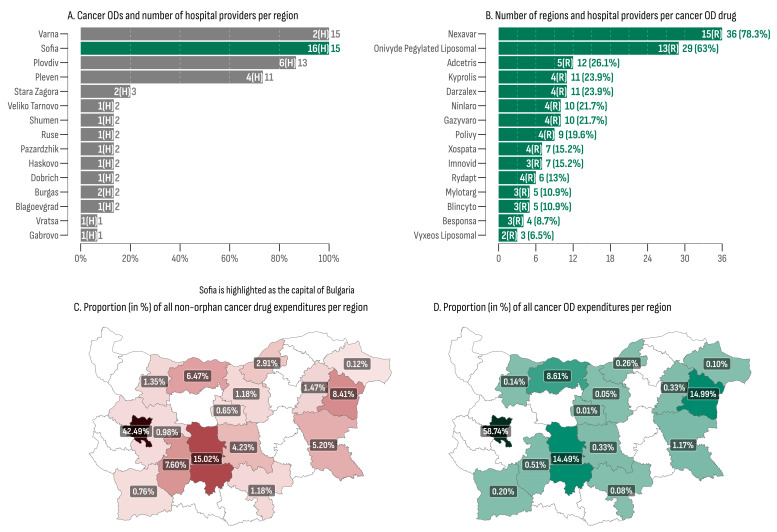
(**A**) Cancer ODs and number of hospital providers per region; (**B**) number of regions and hospital providers per cancer OD drug; (**C**) proportion (in %) of all non-orphan cancer drug expenditures per region; (**D**) proportion (in %) of all cancer OD expenditures per region.

**Figure 4 cancers-16-01489-f004:**
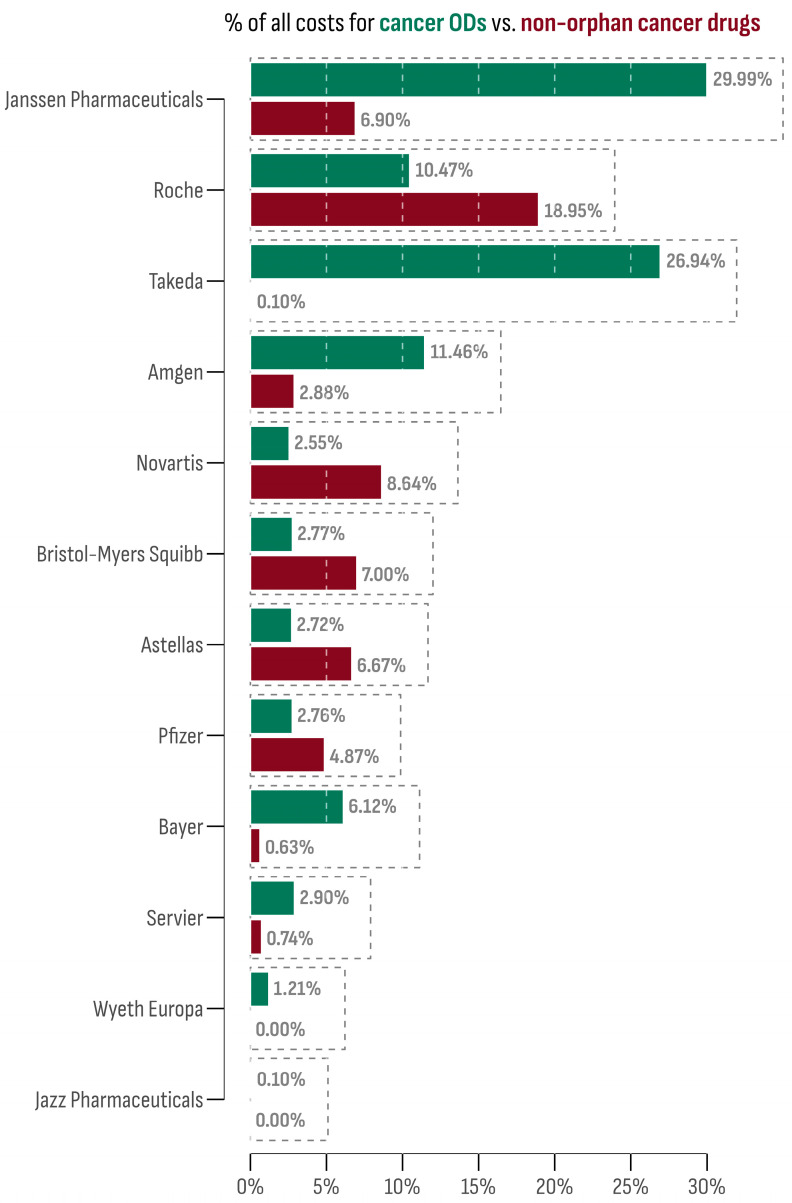
Market share of each company among orphan and non-orphan cancer drug products.

**Table 1 cancers-16-01489-t001:** Data sources and extracted variables.

Data Source	Description	Extracted Variables	Link ^1^
EMA	List of all authorized medicinal products	Product name, ATC code, therapeutic indication, orphan designation status, date of market authorization	https://www.ema.europa.eu/en/medicines/download-medicine-data
NCPRMP	Positive Drug List (PDL) database	Reimbursement status, date of inclusion in the PDL	https://data.egov.bg/resource/download/ba1ab4be-8695-4377-9edf-464233e2fc39/csv
NHIF	Annual budget reports	Total expenses related to all medicinal treatments	https://www.nhif.bg/bg
NHIF	Monthly expenditures for all medicinal treatments	Product (market) name, number of treated patients, ATC, INN, region, hospital, ICD code	https://www.nhif.bg/bg/nzok/medicine/5
EMA	List of all authorized medicinal products	Product name, ATC code, therapeutic indication, orphan designation status, date of market authorization	https://www.ema.europa.eu/en/medicines/download-medicine-data
NCPRMP	Positive Drug List (PDL) database	Reimbursement status, date of inclusion in the PDL	https://data.egov.bg/resource/download/ba1ab4be-8695-4377-9edf-464233e2fc39/csv

^1^ All links were accessed on 3 January 2024.

**Table 2 cancers-16-01489-t002:** List of all 15 cancer ODs accessible in Bulgaria and their therapeutic indications.

Trade Name	INN	Indication
Adcetris	Brentuximab Vedotin	Hodgkin’s lymphoma
Blincyto	Blinatumomab	Precursor Cell Lymphoblastic Leukemia Lymphoma
Darzalex	Daratumumab	Multiple Myeloma
Gazyvaro	Obinutuzumab	Leukemia, Lymphocytic, Chronic, B-Cell
Kyprolis	Carfilzomib	Multiple Myeloma
Nexavar	Sorafenib	Carcinoma, Hepatocellular and Carcinoma, Renal Cell
Ninlaro	Ixazomib	Multiple Myeloma
Rydapt	Midostaurin	Leukemia, Myeloid, Acute and Mastocytosis
Besponsa	Inotuzumab ozogamicin	Precursor Cell Lymphoblastic Leukemia Lymphoma
Imnovid	Pomalidomide	Multiple Myeloma
Mylotarg	Gemtuzumab ozogamicin	Leukemia, Myeloid, Acute
Onivyde Pegylated Liposomal	Irinotecan	Pancreatic Neoplasms
Polivy	Polatuzumab vedotin	Lymphoma, B-Cell
Xospata	Gilteritinib	Leukemia, Myeloid, Acute
Vyxeos Liposomal	Daunorubicin/Cytarabine	Leukemia, Myeloid, Acute

**Table 3 cancers-16-01489-t003:** Total annual and median monthly cancer OD expenditures (July 2020–September 2023).

Indicator/Year	2020	2021	2022	2023
Months	July–December (6)	January–December (12)	January–December (12)	January–September (9)
Total expenditures	7,573,302	18,564,033	22,680,446	25,535,712
% of all cancer drugs expenditures	4.75%	5.17%	5.47%	6.71%
% of NHIF medicinal expenditures	0.24%	2.73%	2.93%	3.77%
Median monthly treated patients	270	287	334	432
Increase in median monthly treated patients Δ%	-	6%	16%	29%
Median monthly cancer OD expenditures (in EUR)	1,267,187	1,487,890	1,910,242	2,884,393
Increase in median monthly cancer OD expenditures Δ%	base	17%	28%	51%
Median monthly expenditure per patient (in EUR)	4701.99	5184.29	5719.29	6676.84

**Table 4 cancers-16-01489-t004:** Median monthly number of patients treated and median monthly cancer OD expenditures in EUR (July 2020–September 2023).

	Year			
Region	2020	2021	2022	2023
Blagoevgrad	3 (9607)	1 (1354)	1 (3043)	1 (2302)
Burgas	14 (46,398)	14 (24,281)	4 (11,419)	3 (6812)
Dobrich	1 (3200)	1 (3190)	2 (1817)	1 (3698)
Gabrovo	1 (1716)	1 (871)	—	—
Haskovo	1 (3202)	2 (3200)	2 (3033)	1 (2980)
Pazardzhik	6 (18,538)	6 (9670)	2 (5093)	2 (3125)
Pleven	23 (107,143)	18 (93,392)	25 (150,814)	53 (309,897)
Plovdiv	38 (169,954)	40 (194,117)	42 (230,639)	62 (432,887)
Ruse	4 (12,809)	3 (5418)	1 (2767)	1 (2767)
Shumen	4 (9608)	4 (5418)	1 (3043)	2 (6452)
Sofia	128 (693,117)	158 (917,017)	202 (1,166,230)	231 (1,568,788)
Stara Zagora	4 (14,338)	4 (7365)	1 (1851)	2 (5011)
Varna	38 (158,481)	38 (188,040)	60 (298,279)	81 (523,459)
Veliko Tarnovo	1 (3143)	1 (1354)	—	1 (1611)
Vratsa	5 (9435)	2 (4720)	1 (97)	3 (6004)

## Data Availability

The complete data repository for this study is available on GitHub https://github.com/kostadinoff/Availability-and-access-to-orphan-drugs-for-rare-cancers-in-Bulgaria (created on 13 March 2024).

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
