# Peer review of "Availability and Access to Orphan Drugs for Rare Cancers in Bulgaria: Analysis of Delays and Public Expenditures"

_cancers, 2024, doi:10.3390/cancers16081489_

Round 1

Reviewer 1 Report

Comments and Suggestions for Authors

It was truly a pleasure to review the manuscript, which clearly reflects the efforts of the authors. I commend the authors for their analysis of the orphan cancer drug market in Bulgaria. One of the main strengths is related to the overall and detailed analysis that the authors performed on the local cancer orphan drug market. All methods and materials to achieve the goal stated are reliable and accurate, explained in details. The authors mention the limitations in a comprehensive manner that meet the Journal guidance. Moreover, all figures express the results and highlight properly the main findings.

However, I would like to raise some comments and address them to the authors:

 1. In further enhancing the manuscript, I suggest the authors to include more relevant citations, which could contribute to the discussion (relevant for the country and the region).

 2. An addition to the discussion section would be an analysis comparing the accessibility of cancer and non-cancer orphan drugs, drawing from existing studies. It would be beneficial to explore the position of orphan cancer drugs in relation to other orphan drugs. Thus, the manuscript could provide an understanding of the broader landscape of orphan drug accessibility and highlight any specific challenges or opportunities unique to the cancer orphan drug market.

3. I recommend the authors to comment in a separate paragraph the advantages and disadvantages of the current legal framework and how it affects access to orphan cancer medicines and whether/how to be improved considering the findings of the current study (to state some recommendations from their point of view as experts).

Reviewer 2 Report

Comments and Suggestions for Authors

Interesting analysis of  OD situation for rare cancers in Bulgaria. The methods applied are in line with standard requirements, The aim of the study is clearly presented. Results were critically provided, and discussion is based on the good data. 

Minor points:

1. Please discuss in more detail the probability of respective  improvements in Bulgaria 

2. Limitations sections: Why do you use firstly rather than first, similar issue for secondly....

Comments on the Quality of English Language

English needs improvement. 
